# BACKPROPAMINE: TRAINING SELF-MODIFYING NEURAL NETWORKS WITH DIFFERENTIABLE NEUROMODULATED PLASTICITY

**Thomas Miconi,**[*] **Aditya Rawal, Jeff Clune & Kenneth O. Stanley**
Uber AI Labs
`tmiconi|aditya.rawal|jeffclune|kstanley@uber.com`

## ABSTRACT

The impressive lifelong learning in animal brains is primarily enabled by plastic changes in synaptic connectivity. Importantly, these changes are not passive, but are actively controlled by neuromodulation, which is itself under the control of the brain. The resulting self-modifying abilities of the brain play an important role in learning and adaptation, and are a major basis for biological reinforcement learning. Here we show for the first time that artificial neural networks with such neuromodulated plasticity can be trained with gradient descent. Extending previous work on differentiable Hebbian plasticity, we propose a differentiable formulation for the neuromodulation of plasticity. We show that neuromodulated plasticity improves the performance of neural networks on both reinforcement learning and supervised learning tasks. In one task, neuromodulated plastic LSTMs with millions of parameters outperform standard LSTMs on a benchmark language modeling task (controlling for the number of parameters). We conclude that differentiable neuromodulation of plasticity offers a powerful new framework for training neural networks.

## 1 INTRODUCTION

Neural networks that deal with temporally extended tasks must be able to store traces of past events. Often this memory of past events is maintained by neural activity reverberating through recurrent connections; other methods for handling temporal information exist, including memory networks (Sukhbaatar et al., 2015) or temporal convolutions (Mishra et al., 2017). However, in nature, the primary basis for long-term learning and memory in the brain is *synaptic plasticity* – the automatic modification of synaptic weights as a function of ongoing activity (Martin et al., 2000; Liu et al., 2012). Plasticity is what enables the brain to store information over the long-term about its environment that would be impossible or impractical for evolution to imprint directly into innate connectivity (e.g. things that are different within each life, such as the language one speaks).

Importantly, these modifications are not a passive process, but are actively modulated on a moment-to-moment basis by dedicated systems and mechanisms: the brain can "decide" where and when to modify its own connectivity, as a function of its inputs and computations. This *neuromodulation* of plasticity, which involves several chemicals (particularly dopamine; Calabresi et al. 2007; He et al. 2015; Li et al. 2003; Yagishita et al. 2014), plays an important role in learning and adaptation (Molina-Luna et al., 2009; Smith-Roe & Kelley, 2000; Kreitzer & Malenka, 2008). By allowing the brain to control its own modification as a function of ongoing states and events, the neuromodulation of plasticity can filter out irrelevant events while selectively incorporating important information, combat catastrophic forgetting of previously acquired knowledge, and implement a self-contained reinforcement learning algorithm by altering its own connectivity in a reward-dependent manner (Schultz et al., 1997; Niv, 2009; Frank et al., 2004; Hoerzer et al., 2014; Miconi, 2017; Ellefsen et al., 2015; Velez & Clune, 2017).

The complex organization of neuromodulated plasticity is not accidental: it results from a long process of evolutionary optimization. Evolution has not only designed the general connection pattern

---

[*]Correspondence: `tmiconi@uber.com`

of the brain, but has also sculpted the machinery that controls neuromodulation, endowing the brain with carefully tuned self-modifying abilities and enabling efficient lifelong learning. In effect, this coupling of evolution and plasticity is a meta-learning process (the original and by far most powerful example of meta-learning), whereby a simple but powerful optimization process (evolution guided by natural selection) discovered how to arrange elementary building blocks to produce remarkably efficient learning agents.

Taking inspiration from nature, several authors have shown that evolutionary algorithms can design small neural networks (on the order of hundreds of connections) with neuromodulated plasticity (see the "Related Work" section below). However, many of the spectacular recent advances in machine learning make use of gradient-based methods (which can directly translate error signals into weight gradients) rather than evolution (which has to discover the gradients through random weight-space exploration). If we could make plastic, neuromodulated networks amenable to gradient descent, we could leverage gradient-based methods for optimizing and studying neuromodulated plastic networks, expanding the abilities of current deep learning architectures to include these important biologically inspired self-modifying abilities.

Here we build on the differentiable plasticity framework (Miconi, 2017; Miconi et al., 2018) to implement differentiable neuromodulated plasticity. As a result, for the first time to our knowledge, we are able to train neuromodulated plastic networks with gradient descent. We call our framework *backpropamine* in reference to its ability to emulate the effects of natural neuromodulators (like dopamine) in artificial neural networks trained by backpropagation. Our experimental results establish that neuromodulated plastic networks outperform both non-plastic and non-modulated plastic networks, both on simple reinforcement learning tasks and on a complex language modeling task involving a multi-million parameter network. By showing that neuromodulated plasticity can be optimized through gradient descent, the backpropamine framework potentially provides more powerful types of neural networks, both recurrent and feedforward, for use in all the myriad domains in which neural networks have had tremendous impact.

## 2 RELATED WORK

Neuromodulated plasticity has long been studied in evolutionary computation. Evolved networks with neuromodulated plasticity were shown to outperform both non-neuromodulated and non-plastic networks in various tasks (e.g. Soltoggio et al. 2008; Risi & Stanley 2012; see Soltoggio et al. 2017 for a review). A key focus of neuromodulation in evolved networks is the mitigation of *catastrophic forgetting*, that is, allowing neural networks to learn new skills without overwriting previously learned skills. By activating plasticity only in the subset of neural weights relevant for the task currently being performed, knowledge stored in other weights about different tasks is left untouched, alleviating catastrophic forgetting (Ellefsen et al., 2015; Velez & Clune, 2017). However, evolved networks were historically relatively small and operated on low-dimensional problem spaces.

The differentiable plasticity framework (Miconi, 2016; Miconi et al., 2018) allows the plasticity of individual synaptic connections to be optimized by gradient descent, in the same way that standard synaptic weights are. However, while it could improve performance in some tasks over recurrence without plasticity, this method only facilitated passive, non-modulated plasticity, in which weight changes occur automatically as a function of pre- and post-synaptic activity. Here we extend this framework to implement differentiable neuromodulated plasticity, in which the plasticity of connections can be modulated moment-to-moment through a signal computed by the network. This extension allows the network itself to decide over its lifetime where and when to be plastic, endowing the network with true self-modifying abilities.

There are other conceivable though more complex approaches for training self-modifying networks. For example, the weight modifications can themselves be computed by a neural network (Schmidhuber, 1993b; Schlag & Schmidhuber, 2017; Munkhdalai & Yu, 2017; Wu et al., 2018). However, none so far have taken the simple approach of directly optimizing the neuromodulation of plasticity itself within a single network, through gradient descent instead of evolution, as investigated here.

## 3 METHODS

### 3.1 BACKGROUND: DIFFERENTIABLE HEBBIAN PLASTICITY

The present work builds upon the existing differentiable plasticity framework (Miconi, 2016; Miconi et al., 2018), which allows gradient descent to optimize not just the weights, but also the plasticity of each connection. In this framework, each connection in the network is augmented with a Hebbian plastic component that grows and decays automatically as a result of ongoing activity. In effect, each connection contains a fixed and a plastic component:

$$x_j(t) = \sigma\Big\{ \sum_{i \in \text{inputs to } j} (w_{i,j} + \alpha_{i,j}\text{Hebb}_{i,j}(t))x_i(t-1)\Big\} \tag{1}$$

$$\text{Hebb}_{i,j}(t+1) = \text{Clip}(\text{Hebb}_{i,j}(t) + \eta x_i(t-1)x_j(t)), \tag{2}$$

where $x_i(t)$ is the output of neuron $i$ at time $t$, $\sigma$ is a nonlinearity (we use $\tanh$ in all experiments), $w_{i,j}$ is the baseline (non-plastic) weight of the connection between neurons $i$ and $j$, and $\alpha_{i,j}$ is the *plasticity coefficient* that scales the magnitude of the plastic component of the connection. The plastic content is represented by the *Hebbian trace* $\text{Hebb}_{i,j}$, which accumulates the product of pre- and post-synaptic activity at connection $i,j$, as shown in Eq. 2.

$\text{Hebb}_{i,j}$ is initialized to zero at the beginning of each episode/lifetime, and is updated automatically according to Eq. 2: it is a purely episodic/intra-life quantity. By contrast, $w_{i,j}$, $\alpha_{i,j}$ and $\eta$ are the structural components of the network, which are optimized by gradient descent between episodes/lifetimes to minimize the expected loss over an episode.

The function $\text{Clip}(x)$ in Eq. 2 is any function or procedure that constrains $\text{Hebb}_{i,j}$ to the $[-1, 1]$ range, to negate the inherent instability of Hebbian learning. In previous work (Miconi et al., 2018), this function was either a simple decay term, or a normalization implementing Oja's rule (Oja, 2008). In the present paper it is simply a hard clip ($x \leftarrow 1$ if $x > 1$; $x \leftarrow -1$ if $x < -1$). Compared to previously used operations, this simple operation turned out to produce equal or superior performance on the tasks in this paper.

Note the distinction between the $\eta$ and $\alpha_{i,j}$ parameters: $\eta$ is the intra-life "learning rate" of plastic connections, which determines how fast new information is incorporated into the plastic component, while $\alpha_{i,j}$ is a scale parameter, which determines the maximum magnitude of the plastic component (since $\text{Hebb}_{i,j}$ is constrained to the [-1,1] range).

Importantly, in contrast to other approaches using *uniform* plasticity (Schmidhuber, 1993a), including "fast weights" (Ba et al., 2016), the amount of plasticity in each connection (represented by $\alpha_{i,j}$) is *trainable*, allowing the meta-optimizer to design complex learning strategies (see Miconi et al. 2018 for a discussion of this point, and experimental comparisons that demonstrate and explain superior performance of differentiable plasticity over uniform-plastic networks).

An important aspect of differentiable plasticity is extreme ease of implementation: implementing a plastic recurrent network only requires less than four additional lines of code over a standard recurrent network implementation (Miconi et al., 2018). The Backpropamine framework described below inherits this simplicity; in particular, the "simple neuromodulation" approach does not require any additional code over differentiable plasticity, but merely a modification of it.

### 3.2 BACKPROPAMINE: DIFFERENTIABLE NEUROMODULATION OF PLASTICITY

Two methods are proposed to introduce neuromodulated plasticity within the differentiable plasticity framework. In both cases, plasticity is modulated on a moment-to-moment basis by a network-controlled neuromodulatory signal $M(t)$. The computation of $M(t)$ could be done in various ways; at present, it is simply a single scalar output of the network, which is used either directly (for the simple RL tasks) or passed through a meta-learned vector of weights (one for each connection, for the language modeling task). We now explain how the equations of differentiable plasticity are modified to make use of this neuromodulatory signal.

### 3.2.1 SIMPLE NEUROMODULATION

The simplest way to introduce neuromodulation of plasticity in this framework is to make the (global) $\eta$ parameter depend on the output of one or more neurons in the network. Because $\eta$ essentially determines the rate of plastic change, placing it under network control allows the network to determine how plastic connections should be at any given time. Thus, the only modification to the equations above in this *simple neuromodulation* variant is to replace $\eta$ in Eq. 2 with the network-computed, time-varying neuromodulatory signal $M(t)$. That is, Eq. 2 is replaced with

$$\text{Hebb}_{i,j}(t+1) = Clip(\text{Hebb}_{i,j}(t) + M(t)x_i(t-1)x_j(t)). \tag{3}$$

### 3.2.2 RETROACTIVE NEUROMODULATION AND ELIGIBILITY TRACES

More complex schemes are possible. In particular, we introduce an alternative neuromodulation scheme that takes inspiration from the short-term retroactive effects of neuromodulatory dopamine on Hebbian plasticity in animal brains. In several experiments, dopamine was shown to retroactively gate the plasticity induced by *past* activity, within a short time window of about 1s (Yagishita et al., 2014; He et al., 2015; Fisher et al., 2017; Cassenaer & Laurent, 2012). Thus, Hebbian plasticity does not directly modify the synaptic weights, but creates a fast-decaying "potential" weight change, which is only incorporated into the actual weights if the synapse receives dopamine within a short time window. As a result, biological Hebbian traces essentially implement a so-called *eligibility trace* (Sutton et al., 1998), keeping memory of which synapses contributed to recent activity, while the dopamine signal modulates the transformation of these eligibility traces into actual plastic changes. Such mechanisms have been modelled in computational neuroscience studies, e.g. (Izhikevich, 2007; Hoerzer et al., 2014; Fiete et al., 2007; Soltoggio & Steil, 2013; Miconi, 2017) (see Gerstner et al. 2018 for a recent review of this concept).

Our framework easily accommodates this more refined model of dopamine effects on plasticity. We simply replace Eq. 2 above with the two equations,

$$\text{Hebb}_{i,j}(t+1) = Clip(\text{Hebb}_{i,j}(t) + M(t)E_{i,j}(t)) \tag{4}$$

$$E_{i,j}(t+1) = (1-\eta)E_{i,j}(t) + \eta x_i(t-1)x_j(t). \tag{5}$$

Here $E_{i,j}(t)$ (the eligibility trace at connection $i,j$) is a simple exponential average of the Hebbian product of pre- and post-synaptic activity, with trainable decay factor $\eta$. $\text{Hebb}_{i,j}(t)$, the actual plastic component of the connection (see Eq. 1), simply accumulates this trace, but gated by the current value of the dopamine signal $M(t)$. Note that $M(t)$ can be positive or negative, approximating the effects of both rises and dips in the baseline dopamine levels (Schultz et al., 1997).

## 4 EXPERIMENTS

### 4.1 TASK 1: CUE-REWARD ASSOCIATION

Our first test task is a simple meta-learning problem that emulates an animal behavioral learning task, as described in Figure 4.1 (Left). In each episode, one of four input cues is arbitrarily chosen as the *Target* cue. Repeatedly, the agent is shown two cues in succession, randomly chosen from the possible four, then a *Response* cue during which the agent must respond 1 if the Target cue was part of the pair, or 0 otherwise. A correct response produces a reward of 1.0, while an incorrect response returns reward -1.0 (this is a two-alternative forced choice task: a response of either 1 or 0 is always produced). This process iterates for the duration of the episode, which is 200 time steps. The cues are binary vectors of 20 bits, randomly generated at the beginning of each episode. To prevent simple time-locked scheduling strategies, a variable number of zero-input time steps are randomly inserted, including at least one after each presentation of the Go cue; as a result, the length of each trial varies, and the number of trials per episode is somewhat variable (the mean number of trials per episode is 15).

The architecture is a simple recurrent network with 200 neurons in the hidden recurrent layer. Only the recurrent layer is plastic: input and output weights are non-plastic, having only $w_{i,j}$ coefficients.

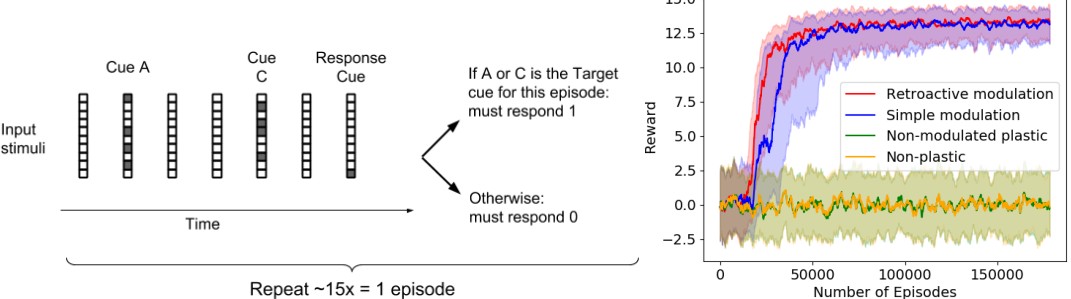

Figure 1: Left: Description of the task. Right: Training curves for the cue-reward association task (medians and inter-quartile ranges of rewards per episode over 10 runs). Modulated plastic networks (red, blue) learn the task, while non-modulated and non-plastic networks (green, orange) fail.

There are 24 inputs: 20 binary inputs for the current cue and one input providing the time elapsed since the start of the episode, as well as two binary inputs for the one-hot encoded response at the previous time step and one real-valued channel for the reward received at the previous time step, in accordance with common meta-learning practice (Wang et al., 2016; Duan et al., 2016). There are four outputs: two binary outputs for the one-hot encoded response, plus an output neuron that predicts the sum of future discounted rewards $V(t)$ over the remainder of the episode (as mandated by the A2C algorithm that we use for meta-training, following Wang et al. (2016)), and the neuromodulatory signal $M(t)$. The two response outputs undergo a softmax operation to produce probabilities over the response, while the $M(t)$ signal is passed through a $\tanh$ nonlinearity and the $V(t)$ output is a pure linear output. All gradients are clipped at norm 7.0, which greatly improved stability.

Training curves are shown in Figure 4.1 (Right; each curve shows the median and inter-quartile range over 10 runs). Neuromodulatory approaches succeed in learning the task, while non-neuromodulatory networks (Miconi, 2016; Miconi et al., 2018)) and non-plastic, simple recurrent networks fail to learn it. We hypothesize that this dramatic difference is related to the relatively high dimensionality of the input cues: just as non-modulated plastic networks seemed to outperform non-plastic networks specifically when required to memorize arbitrary high-dimensional stimuli (Miconi et al., 2018), neuromodulation seems to specifically help memorizing reward associations with such arbitrary high-dimensional stimuli (see Appendix).

To illustrate the behavior of neuromodulation, we plotted the output of the neuromodulator neuron for several trials. These graphs reveal that neuromodulation reacts to reward in a complex, time-dependent manner (see Appendix).

## 4.2    TASK 2: MAZE NAVIGATION TASK

For a more challenging problem, we also tested the approach on the grid maze exploration task introduced by Miconi et al. (2018). Here, the maze is composed of $9 \times 9$ squares, surrounded by walls, in which every other square (in either direction) is occupied by a wall. Thus the maze contains 16 wall squares, arranged in a regular grid except for the center square (Figure 2, left). The shape of the maze is fixed and unchanging over the whole task. At each episode, one non-wall square is randomly chosen as the reward location. When the agent hits this location, it receives a reward and is immediately transported to a *random* location in the maze. Each episode lasts 200 time steps, during which the agent must accumulate as much reward as possible. The reward location is fixed within an episode and randomized across episodes. Note that the reward is invisible to the agent, and thus the agent only knows it has hit the reward location by the activation of the reward input at the next step (and possibly by the teleportation, if it can detect it).

The architecture is the same as for the previous task, but with only 100 recurrent neurons. The outputs consist of 4 action channels (i.e. one for each of the possible actions: left, right, up or down) passed through a softmax, as well as the pure linear $V(t)$ output and the $M(t)$ neuromodulatory signal passed through a $\tanh$ nonlinearity. Inputs to the agent consist of a binary vector describing the $3 \times 3$

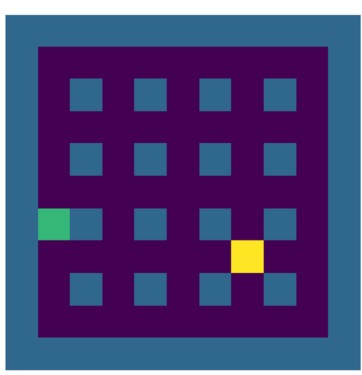 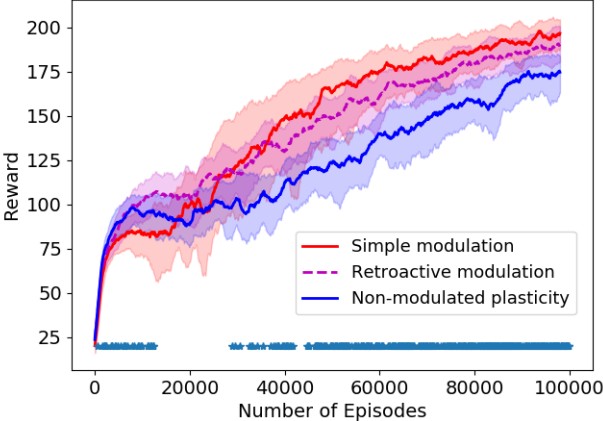

Figure 2: Maze navigation task. Left: layout of the maze, including an example agent location (yellow) and reward location (green, for illustration only: the reward is not visible to the agent). Right: Training curves for the maze exploration task: median and inter-quartile range of reward over 9 runs for each episode. Cyan stars (bottom) indicate statistically significant difference between simple neuromodulation and non-modulated plasticity at $p < 0.05$ (Wilcoxon rank-sum test).

neighborhood centered on the agent (each element being set to 1 or 0 if the corresponding square is or is not a wall), plus four additional inputs for the one-hot encoded action taken at the previous time step, and one input for the reward received at the previous time step, following common practice (Wang et al., 2016). Again, only recurrent weights are plastic: input-to-recurrent and recurrent-to-output weights are non-plastic. Results in Figure 2 show that modulatory approaches again outperform non-modulated plasticity.

### 4.3    TASK 3: LANGUAGE MODELING

Word-level language modeling is a supervised learning sequence problem, where the goal is to predict the next word in a large language corpus. Language modeling requires storing long term context, and therefore LSTM models (Hochreiter & Schmidhuber, 1997) generally perform well on this task (Zaremba et al., 2014). The goal of this experiment is to study the benefits of adding plasticity and neuromodulation to LSTMs.

The Penn Tree Bank corpus (PTB), a well known benchmark for language modeling (Marcus et al., 1993), is used here for comparing different models. The dataset consists of 929k training words, 73k validation words, and 82k, test words, with a vocabulary of 10k words.

For this task, we implemented neuromodulated plasticity in two different models: a basic model with 4.8 million parameters inspired from Zaremba et al. (2014), and a much larger and more complex model with 24.2 million parameters, forked from Merity & Socher (2017). The smaller model allowed for more experimentation, while the larger model showcases the results of neuromodulation on a complex model with (at the time of writing) state-of-the-art performance.

Detailed experimental descriptions are provided in the Appendix, summarized here: For the basic model, each network consists of an embedding layer, followed by two LSTM layers (approximately of size 200). The size of the LSTM layers is adjusted to ensure that the total number of trainable parameters remains constant across all experiments; note that this includes all plasticity-related additional parameters, i.e. $\alpha_{i,j}$ as well as the additional parameters related to neuromodulation (see Appendix). The final layer is a softmax layer of size 10k. The network is unrolled for 20 time steps during backpropagation through time (Werbos, 1990). The norm of the gradient is clipped at 5. This setup is similar to the non-regularized model described by Zaremba et al. (2014). One difference is that an extra L2 penalty is added to the weights of the network here (adding this penalty consistently improves results for all the models).

Table 1: Test Perplexity Results on Penn-Tree Bank. Lower values are better. Basic model: mean and 95% CI over 16 runs. Large model: median and (min, max) over 5 runs.

| Model | Test Perplexity |
|---|---|
| Baseline LSTM (similar to Zaremba et al. (2014)) | $104.26 \pm 0.22$ |
| LSTM with Differential Plasticity | $103.80 \pm 0.25$ |
| LSTM with Simple Neuromodulation | $102.65 \pm 0.30$ |
| LSTM with Retroactive Neuromodulation | $102.48 \pm 0.28$ |
| Baseline large LSTM model (from Merity & Socher (2017)) | 62.48 (62.40, 62.60) |
| Large LSTM model with neuromodulated plasticity | 61.44 (61.37, 61.68) |

The large model, as described in (Merity & Socher, 2017), consists of an embedding of size 400 followed by three LSTMs with 1150, 1150 and 400 cells respectively (which we reduce to 1149, 1149 and 400 for the plastic version only to ensure the total number of trainable parameters is not higher). Importantly Merity & Socher (2017) use various regularization techniques. For example, the training process involves switching from standard SGD to Averaged-SGD after a number of epochs. The main departures from Merity & Socher (2017) is that we do not implement recurrent dropout (feedforward dropout is preserved) and reduce batch size to 7 due to computational limitations. Other hyperparameters are taken "as is" without any tuning. See Appendix for details.

Four versions of the smaller model are evaluated here (Table 1). (1) The *Baseline LSTM* model (described in the previous paragraph)[1]. (2) *LSTM with Differentiable Plasticity*: there are four recurrent connections in each LSTM node and here, plasticity is added to one of them (see A.1 for details) as per equations 1 and 2. Because the number of plastic connections is large, each plastic connection has its own individual $\eta$ so that their values can be individually tuned by backpropagation. (3) *LSTM with Simple Neuromodulation:* here simple neuromodulation is introduced following equation 3. The $\eta$ parameters are replaced by the output of a neuron $M(t)$. $M(t)$ itself receives as input a weighted combination of the hidden layer's activations, where the weights are learned in the usual way. There is one $M(t)$ associated with each LSTM layer. (4) *LSTM with Retroactive Neuromodulation*: this model is the same as the LSTM with Simple Neuromodulation, except it uses the equations that enable eligibility traces (equations 4 and 5). Additional details for the plastic and neuromodulated plastic LSTMs are described in the Appendix.

For each of the four models, we separately searched for the best hyperparameters with equally-powered grid-search. Each model was then run 16 times with its best hyperparameter settings. The mean test perplexity of these 16 runs along with the 95% confidence interval is presented in Table 1. Results show that adding differentiable plasticity to LSTM provides slightly, but statistically significantly better results than the Baseline LSTM (Wilcoxon rank-sum test, $p = 0.0044$). Adding neuromodulation further (and statistically significantly) lowers the perplexity over and above the LSTM with differential plasticity ($p = 1e - 6$). Overall, retroactive neuromodulation provides about 1.7 perplexity improvement vs. the Baseline LSTM (statistically significant, $p = 1e - 7$). Retroactive neuromodulation (i.e. with eligibility traces) does outperform simple neuromodulation, but the improvement is just barely not statistically significant at the traditional $p < 0.05$ cutoff ($p = 0.066$). Note that while these figures are far from state-of-the-art results (which use considerably larger, more complex architectures), they all still outperform published work using similar architectures (Zaremba et al., 2014).

For the larger model, we compare a version in which the core LSTM module has been reimplemented to have neuromodulated plastic connections (simple neuromdulation only; no retroactive modulation was implemented), and a baseline model that uses the same LSTM reimplementation but without the plasticity and modulation, in order to make the comparison as equal as possible. Note that in this model, plasticity coefficients are attributed "per neuron": there is only one $\alpha_i$ for each neuron $i$ (as opposed to one per connection), which is applied to all the Hebbian traces of the connections incoming to this neuron. This helps limit the total number of parameter. See the Appendix for a more

---

[1]The Baseline LSTM performance is better than the one published in (Zaremba et al., 2014) due to our hyperparameter tuning, as described in the Appendix.

complete description. The modulated plastic model shows a small improvement over the non-plastic version (Table 1), confirming the results obtained with the smaller model.

## 5 DISCUSSION AND FUTURE WORK

This paper introduces a biologically-inspired method for training networks to self-modify their weights. Building upon the differentiable plasticity framework, which already improved performance (sometimes dramatically) over non-plastic architectures on various supervised and RL tasks (Miconi, 2016; Miconi et al., 2018), here we introduce neuromodulated plasticity to let the network control its own weight changes. As a result, for the first time, neuromodulated plastic networks can be trained with gradient descent, opening up a new research direction into optimizing large-scale self-modifying neural networks.

As a complement to the benefits in the simple RL domains investigated, our finding that plastic and neuromodulated LSTMs outperform standard LSTMs on a benchmark language modeling task (importantly, a central domain of application of LSTMs) is potentially of great importance. LSTMs are used in real-world applications with massive academic and economic impact. Therefore, if plasticity and neuromodulation consistently improve LSTM performance (for a fixed search space size), the potential benefits could be considerable. We intend to pursue this line of investigation and test plastic LSTMs (both neuromodulated and non) on other problems for which LSTMs are commonly used, such as forecasting.

Conceptually, an important comparison point is the "Learning to Reinforcement Learn" (L2RL) framework introduced by Wang et al. (2016; 2018). In this meta-learning framework, the weights do not change during episodes: all within-episode learning occurs through updates to the activity state of the network. This framework is explicitly described (Wang et al., 2018) as a model of the slow sculpting of prefrontal cortex by the reward-based dopamine system, an analogy facilitated by the features of the A2C algorithm used for meta-training (such as the use of a value signal and modulation of weight changes by a reward prediction error). As described in the RL experiments above, our approach adds more flexibility to this model by allowing the system to store state information with weight changes, in addition to hidden state changes. However, because our framework allows the network to update its own connectivity, we might potentially extend the L2RL model one level higher: rather than using A2C as a hand-designed reward-based weight-modification scheme, the system could now determine its own arbitrary weight-modification scheme, which might make use of any signal it can compute (reward predictions, surprise, saliency, etc.) This emergent weight-modifying algorithm (designed over many episodes/lifetimes by the "outer loop" meta-training algorithm) might in turn sculpt network connectivity to implement the meta-learning process described by Wang et al. (2018). Importantly, this additional level of learning (or "meta-meta-learning") is not just a pure flight of fancy: it has undoubtedly taken place in evolution. Because humans (and other animals) can perform meta-learning ("learning-to-learn") during their lifetime (Harlow, 1949; Wang et al., 2018), and because humans are themselves the result of an optimization process (evolution), then meta-meta-learning has not only occurred, but may be the key to some of the most advanced human mental functions. Our framework opens the tantalizing possibility of studying this process, while allowing us to replace evolution with any gradient-based method in the outermost optimization loop.

To investigate the full potential of our approach, the framework described above requires several improvements. These include: implementing multiple neuromodulatory signals (each with their own inputs and outputs), as seems to be the case in the brain (Lammel et al., 2014; Howe & Dombeck, 2016; Saunders et al., 2018); introducing more complex tasks that could make full use of the flexibility of the framework, including the eligibility traces afforded by retroactive modulation and the several levels of learning mentioned above; and addressing the pitfalls in the implementation of reinforcement learning with reward-modulated Hebbian plasticity (e.g. the inherent interference between the unsupervised component of Hebbian learning and reward-based modifications; Frémaux et al. 2010; Frémaux & Gerstner 2015), so as to facilitate the automatic design of efficient, self-contained reinforcement learning systems. Finally, it might be necessary to allow the meta-training algorithm to design the overall architecture of the system, rather than simply the parameters of a fixed, hand-designed architecture. With such a rich potential for extension, our framework for neuromodulated plastic networks opens many avenues of exciting research.

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

# A APPENDIX

## A.1 PLASTIC LSTMs: BASIC MODEL

### A.1.1 ADDING PLASTICITY TO LSTMs

Each LSTM node consists of four weighted recurrent paths through $i_t$, $j_t$, $f_t$ and $o_t$ as shown in the equations below:

$$i_t = \tanh(W_{xi}x_t + W_{hi}h_{t-1} + b_i) \tag{6}$$

$$j_t = \sigma(W_{xj}x_t + W_{hj}h_{t-1} + b_j) \tag{7}$$

$$f_t = \sigma(W_{xf}x_t + W_{hf}h_{t-1} + b_f) \tag{8}$$

$$o_t = \sigma(W_{xo}x_t + W_{ho}h_{t-1} + b_o) \tag{9}$$

$$c_t = f_t \otimes c_{t-1} + i_t \otimes j_t \tag{10}$$

$$h_t = \tanh(c_t) \otimes o_t \tag{11}$$

$j_t$, $f_t$ and $o_t$ are used for controlling the data-flow through the LSTM and $i_t$ is the actual data. Therefore, plasticity is introduced in the path that goes through $i_t$ (adding plasticity to the control paths of LSTM is for future-work) . The corresponding pre-synaptic and post-synaptic activations (denoted by $x_i(t-1)$ and $x_j(t)$ respectively in equations 1 and 2) are $h_{t-1}$ and $i_t$. A layer of size 200 has 40k (200×200) plastic connections. Each plastic connection has its own individual $\eta$ (used in equation 2) that is learned through backpropagation. The plasticity coefficients ($\alpha_{i,j}$) are used as shown in equation 1.

### A.1.2 Adding neuromodulation to LSTMs

As shown in equation 3, for simple neuromodulation, the $\eta$ is replaced by the output of a network computed neuron $M(t)$. For neuromodulated LSTMs, individual $\eta$ for each plastic connection is replaced by the output of a neuron ($M(t)$) that has a fan-out equal to the number of plastic connections. The input to this neuron is the activations $h_{t-1}$ of the layer from the previous time-step. Each LSTM layer has its dedicated neuromodulatory neuron. Other variations of this setting include having one dedicated neuromodulatory neuron per node or having one neuromodulatory neuron for the whole network. Preliminary experiments showed that these variations performed worse and therefore they were not further evaluated.

### A.2 More details for Language Modeling experiment

All the four models presented in Table 1 are trained using SGD. Initial learning rate was set 1.0. Each model is trained for 13 epochs. The hidden states of LSTM are initialized to zero; the final hidden states of the current minibatch are used as the initial hidden states of the subsequent minibatch.

Grid-search was performed for four hyperparameters: (1) Learning rate decay factor in the range 0.25 to 0.4 in steps of 0.01. (2) Epoch at which learning rate decay begins in the range - $\{4, 5, 6\}$. (3) Initial scale of weights in the range - $\{0.09, 0.1, 0.11, 0.12\}$. (4) L2 penalty constant in the range - $\{1e-2, 1e-3, 1e-4, 1e-5, 1e-6\}$.

### A.3 Large word-modelling network

In addition to the previous model, we also applied the Backpropamine framework to the much larger, state-of-the-art model described by Merity & Socher (2017). This model consists of three stacked LSTMs with 115, 1150 and 400 cells each, with an input embedding of size 400 and an output softmax layer that shares weights with the input embedding. The model makes use of numerous optimization and regularization techniques. Connections between successive LSTMs implement "variational" dropout, in which a common dropout mask is used for the entire forward and backward pass Gal & Ghahramani (2016). Backpropagation through time uses a variable horizon centered on 70 words. After 45 epochs, the optimizer switches from SGD (without momentum) to Averaged-SGD, which consists in computing standard SGD steps but taking the average of the resulting successive updated weight vectors. This is all in accordance with Merity & Socher (2017). The only differences are that we do not implement weight-dropout in recurrent connections, force the switch to ASGD at 45 epochs for all runs of all models, and limit batch size to 7 due to computational restrictions.

Plasticity coefficients are attributed "per neuron": rather than having and independent $\alpha_{i,j}$ for each connection, each neuron $i$ has a plasticity coefficient $\alpha_i$ that is applied to all its incoming connection (note that Hebbian traces $\text{Hebb}_{i,j}$ are still individually maintained for each connection). This reduces the number of trainable parameters, since $\alpha$ is now a vector of length $N$ rather than a matrix of size $N \times N$ (where $N$ is the number of recurrent neurons).

We implement simple neuromodulation as described in Equation 3. A single neuromodulator neuron with $\tanh$ nonlinearity receives input from all recurrent neurons. This neuromodulator input is then passed through a vector of weights, one per neuron, to produce a different $\eta_i$ for each neuron. In other

words, different neurons $i$ have different $\eta_i$, but these are all fixed multiples of a common value. This is an intermediate solution between having a single $\eta(t)$ for the whole network, and independently computing a separate $\eta_i$ for each neuron, each with its own input weights (which would require $N \times N$ weights, rather than $2 \times N$ for the current solution). Neuromodulation is computed separately to each of the three LSTMs in the model.

For the non-plastic network, the total number of trainable parameters is 24 221 600. For the neuro-modulated plastic version, we reduce the number of hidden cells in LSTMs from 1150 to 1149, which suffices to bring the total number of parameters down to 24 198 893 trainable parameters (rather than 24 229 703 for 1150-cell LSTMs).

All other hyperparameters are taken from Merity & Socher (2017), using the instructions provided on the code repository for their model, available at `https://github.com/salesforce/awd-lstm-lm`. We did not perform any hyperparameter tuning due to computational constraints.

## A.4 DYNAMICS OF NEUROMODULATION

To illustrate the behavior of neuromodulation, we plot the output of the neuromodulator neuron for random trials from several runs of Task 1 (Figure 3). All runs are from well-trained, highly successful network, as seen by the low proportion of negative rewards. For each run, we plot both the value of the neuromodulator output at each time step, and the reward being currently perceived by the network (i.e. the one produced by the response at the previous time step).

The plots reveal rich, complex dynamics that vary greatly between runs. The modulator neuron clearly reacts to reward; however, this reaction is complex, time-dependent and varies from run to run. The topmost run for retroactive modulation tends to produce negative neuromodulation in response to positive reward, and vice-versa; while the second-to-last run for simple neuromodulation tends to to the opposite. A common pattern is to produce negative neuromdulation on the time step just following reward perception (especially for simple neuromodulation). Two of the runs for retroactive modulation exhibit a pattern where reward perception is followed by highly positive, then highly negative neuromodulation. Understanding the mechanism by which these complex dynamics perform efficient within-episode learning is an important direction for future work.

## A.5 CUE-REWARD ASSOCIATION TASK

In the cue-reward association learning task described above, neuromodulated plasticity was able to learn a task that non-modulated plasticity simply could not. What might be the source of this difference? In a previous experiment, we implemented the same task, but using only four fixed 4-bit binary cues for the entire task, namely, '1000', '0100', '0010' and '0001'. In this simplified version of the task, there is no need to memorize the cues for each episode, and the only thing to be learned for each episode is which of the four known cues is associated with reward. This is in contrast with the version used in the paper above, in which the cues are arbitrary 20-bits vectors randomly generated for each episode. With the fixed, four-bit cues, non-modulated plasticity was able to learn the task, though somewhat more slowly than neuromodulated plasticity (see Figure 4).

This suggests neuromodulated plasticity could have a stronger advantage over non-modulated plasticity specifically in situations where the association to be learned involves arbitrary high-dimensional cues, which must be memorized jointly with the association itself. This echoes the results of Miconi et al. (2018), who suggest that plastic networks outperform non-plastic ones specifically on tasks requiring the fast memorization of high-dimensional inputs (e.g. image memorization and reconstruction task in (Miconi et al., 2018)).

Clearly, more work is needed to investigate which problems benefit most from neuromodulated plasticity, over non-modulated or non-plastic approaches. We intend to pursue this line of research in future work.

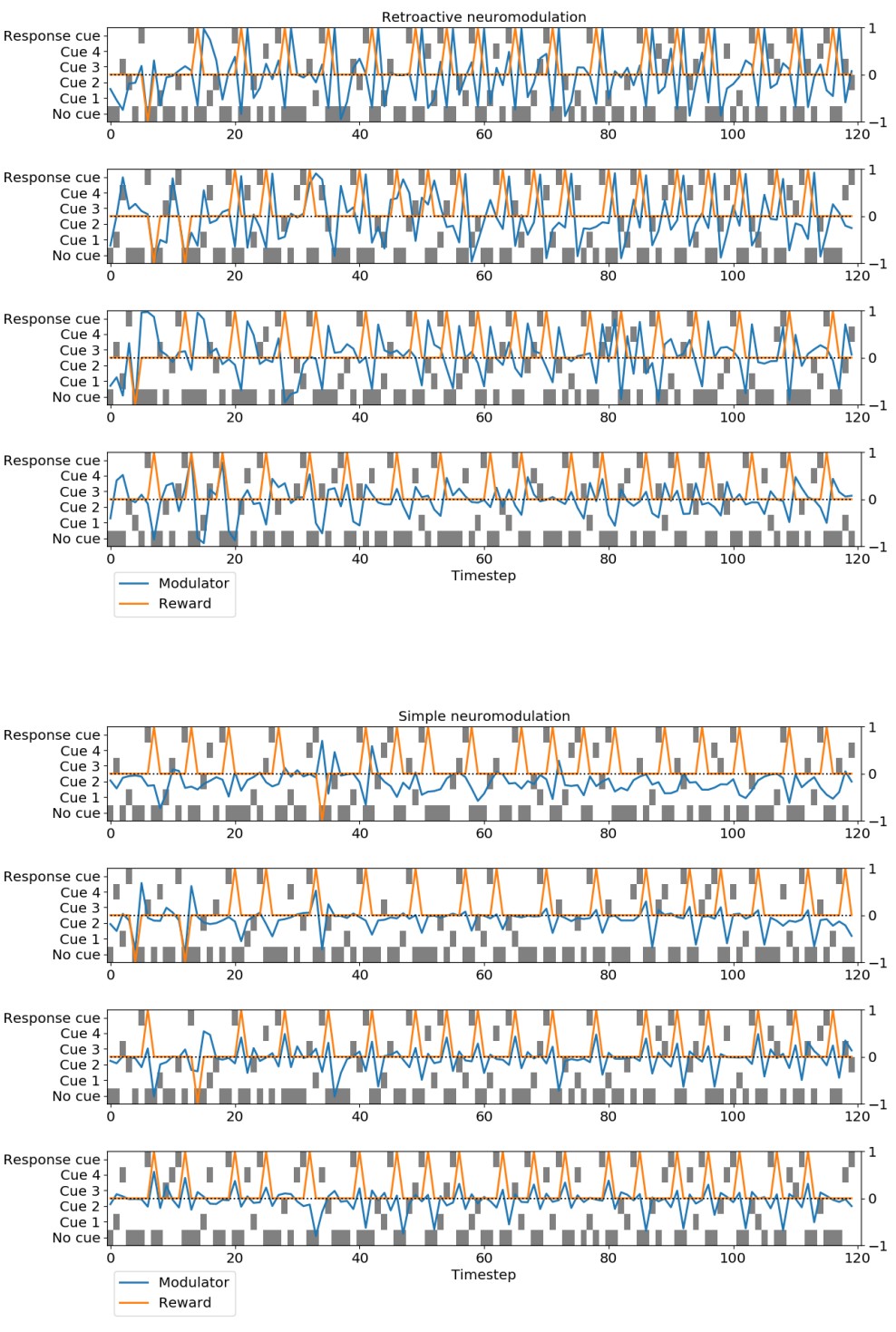

Figure 3: Dynamics of neuromodulation. For both simple and retroactive modulation, we show one trial from each of 4 runs. At each time step, gray squares indicate which cue was shown; orange curves indicates reward at the previous time step (i.e. the one currently fed to the network), which is always -1, 0 or 1; and blue curves indicates the value of the modulator output at the current time step. Notice the diversity of dynamics.

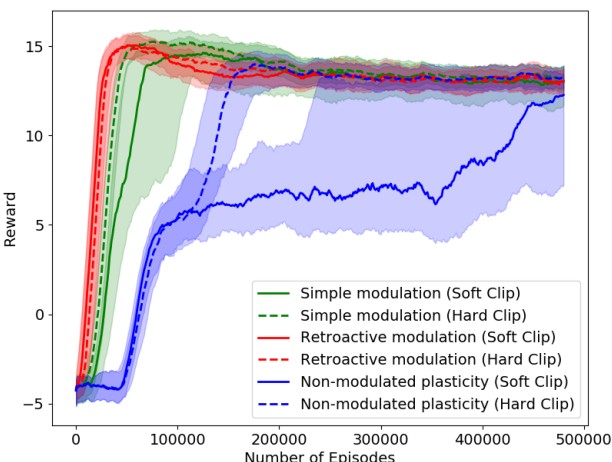

Figure 4: Training curves for the cue-reward association task with fixed, binary four-bit cues (medians and inter-quartile ranges of rewards per episode over 10 runs). "Soft clip" refers to a different clipping operation used in Equation 2; "Hard clip" is the same as used in the present paper, i.e. the simple clipping described in Methods. Note that non-modulated plastic network succeed in solving this task.

