# OpenReview forum: "Backpropamine: training self-modifying neural networks with differentiable neuromodulated plasticity"
_ICLR.cc/2019/Conference_

### Official Review · AnonReviewer2 · 2018-10-15
**Interesting extension of differentiable plasticity with evaluation that falls too short**

**Rating:** 5
**Confidence:** 4

**Review:**

The paper extends previous work on differentiable placticity to include neuro modulation by parameterizing the learning rate of Hebbs update rule. In addition, the authors introduce retroactive modulation that basically allows the system to delay incorporation of plasticity updates via so eligibility traces. Experiments are performaed on 2 simple toy datasets and a simple language modeling task. A newly developed cue-reward association task shows the clear limitations of basic plasticity and how modulation can resolve this. Slight improvements can also be seen on a simple maze navigation task as well as on a basic language modeling dataset.

Overall I like the motivation, provided background information and simplicity of the approach. Furthermore, the cue-reward experiment seems to be a well designed show case for neuro-modulation. However, as the authors acknowledge the overall simplicity of the tasks being evaluated with mostly marginal improvements makes the overall evaluation fall short. Unfortunately the paper doesn't provide any qualitative analysis on how modulation is employed by the models after training. Therefore, although I would like to see an extended version of this paper at the conference, without further experiments and analysis I see the current version rather as an interesting workshop contribution.


Strengths:
- motivation: the natural extension of previous work on differentiable plasticity based on existing knowledge from neuro science is an important next step
- cue reward experiment exemplifies limitations of current plasticity approaches and clearly shows the potential benefits of neuro modulation
- maze navigation shows incremental benefits over non-modulated plasticity
- thorough experimentation
- clipping-trick is a neat observation


Weaknesses:
- evaluation: only on toy tasks (which includes PTB), no real world tasks
- very incremental improvements on PTB over a very simple baseline (far from SotA)
- evaluated models (feed-forward NNs and LSTMs) are very basic and far from current SotA architectures
- no qualitative analysis on how modulation is actually use by the systems. E.g., when is modulation strong and when is it not used


Comments:
- perplexity improvements of less than 1.3 points over plasticity alone (which is the actual baseline for this paper) can hardy be called "significant". Even though they might be statistically significant (meaning nothing more than the two models being statistically different), minor architectural changes can lead to such improvements. Furthermore PTB is not a "challenging" LM benchmark.

---

> ### Author Response · Authors · 2018-11-27
> **Response to Reviewer 3**
>
> Thank you to Reviewer 3 for your thoughtful critique and we are happy that you share our enthusiasm for the motivation behind our approach.   We share your curiosity on the qualitative behavior of such systems, and as documented in this response we have augmented the paper to address that and other of your suggestions.
>
> Re: "- no qualitative analysis on how modulation is actually use by the systems. E.g., when is modulation strong and when is it not used "
>
> Following the reviewer’s suggestion, we have added a figure that shows the dynamics of neuromodulation in the cue-response task (Figure 3, in the Appendix). This figure shows that while neuromodulation clearly reacts to reward, this reaction is complex and varies both within each episode and between runs.
>
> Re: "- perplexity improvements of less than 1.3 points over plasticity alone (which is the actual baseline for this paper) can hardy be called "significant". Even though they might be statistically significant (meaning nothing more than the two models being statistically different), minor architectural changes can lead to such improvements. Furthermore PTB is not a "challenging" LM benchmark."
>
> We agree that, while the differences are statistically significant, they are minor. We were using that word technically, but do not want to give the wrong impression. We have thus modified the text to make it clear that we mean “statistically significant” only. We also removed the adjective “challenging” as regards PTB.
>
> We agree that, ideally, a comparison with SOTA architectures would be desirable. As explained in the response to Reviewer 1, despite all our efforts, we found the technical challenges insurmountable given our computational and engineering resources. We will keep trying to investigate such massive architectures in the future.
>
>
> Importantly, our purpose in this task is to show that, **all other things being equal**, a neuromodulated plastic LSTM can outperform a standard LSTM in realistic settings. We believe that outperforming standard LSTMs (again, all else being equal) on their “workhorse” task domain (language processing) is worthy of notice, especially given the ease of implementation of our method which requires only adding a few lines of codes (<10) to a standard LSTM implementation and can then be used as a drop-in replacement to standard LSTM.

---

> > ### Comment · AnonReviewer2 · 2018-11-27
> > **Re: Rebuttal**
> >
> > I appreciate the addition of the qualitative analysis. However, much like reviewer 1 my main problem with this work lie in the poor evaluation setup. Unfortunately, still all of the tasks are toy tasks and for PTB the results are particularly low. It is hard to trust any improvements in the perplexity regimes reported here, since 10-20 perplexity point gains are easily achievable with simple LSTMs. Given the relatively simple architectural additions made over previous work I would expect a more rigorous evaluation with more experiments and models that can really show the benefit of this idea in more realistic settings. In its current state I see this work as an interesting workshop addition.

---

> > > ### Author Response · Authors · 2018-12-05
> > > **Response and additional experiments.**
> > >
> > > Thank you for your response. In response to reviewer's suggestions, we have performed additional experiment on more complex architectures. In all cases, modulated plasticity provided a moderate but consistent improvement for the same or lower number of parameters. Please see response to Reviewer 1 (who had largely similar comments) for details.

---

> > > > ### Comment · AnonReviewer2 · 2018-12-05
> > > > **RE**
> > > >
> > > > I appreciate the careful evaluation on PTB but unfortunately this doesn't address my main concerns, which are that this is still a toy dataset imo, and that there is no visible advantage of using neuromodulation. What are the error bars on these results? Are the changes significant? The modest improvements can have many different reasons, like the intrinsic regularization from scaling weights, or pure chance given that it was only 1 run. I am not willing to trust improvements on a dataset like PTB if they are not quite large. Ultimately, I strongly believe that LM is not a good ask for the proposed approach and I strongly advise to find better applications and harder datasets for this kind of research. For instance, I believe that neuro-modulation could be beneficial for domain adaptation or few shot learning. So I am sticking with my evaluation of this work and strongly encourage to conduct further and more rigorous experimental work.

---

### Official Review · AnonReviewer3 · 2018-10-29
**Meta-learning to dynamically set the plasticity learning rate**

**Rating:** 9
**Confidence:** 4

**Review:**

Paper summary - This paper extends the differentiable plasticity framework of Miconi et al. (2018) by dynamically modulating the plasticity learning rate. This is accomplished via an output unit of the network which defines the plasticity learning rate for the next timestep. A variation on this dynamic learning rate related to eligibility traces is also proposed.

Both dynamic modulation variations strikingly outperform non-plastic and plastic non-modulated recurrent networks on a cue-reward association task with high-dimensional cues. The methods marginally outperform plastic non-modulated recurrent networks on a 9x9 water maze task. Finally, the authors show that adding dynamic plasticity to a small LSTM without dropout improves performance on Penn Treebank.

The paper motivates dynamic plasticity by analogy to the hypothesized role of dopamine in reward-driven learning in humans and animals.

Clarity -  The paper is very clear and well written. The introduction provides useful insights, motivates the work convincingly, and provides interesting connections to past work.

Originality - I don't know of any other work that models the role of dopamine in quite this way, or that applies dynamic plasticity modulation in settings like these.

Quality - The experiments are well chosen and seem technically sound.

Significance - The results show that meta-learning by gradient descent to modulate the plasticity learning rate is a promising direction -- a significant contribution in my view.

Other Comments - The citation to Zaremba et al. in Table 1 made it seem like the perplexity result on that line of the table was directly from Zaremba et al's paper. I'd recommend removing the citation from that line to avoid confusion.

One thing I would have loved to see from this paper is a comparison of modulated-plasticity LSTMs with the sota from Melis et al., 2017. I gather that Experiment 3 presents small LSTMs without recurrent dropout instead because combining plasticity and dropout proved challenging (or at least the authors haven't tried it yet). I think the paper is solid as-is; positive results in this comparison would take it to the next level.

Questions:
Why were zero-sequences necessary in Experiment 1? This aspect of the task seems somewhat contrived, and it makes me wonder whether the striking failure of the non-modulated RNNs depends on this detail. Perhaps the authors could clarify on what a confounding "time-locked scheduling strategy" would look like in this task?
Why does Experiment 1 present pairs of stimuli, rather than high-dimensional individual stimuli?
Why is non-plastic rnn left out of Figure 2b?

Typos
"However, in Nature," -- no caps
in appendix: "(see Figure A.4)" -- the figure is labeled "Figure 3"

---

> ### Author Response · Authors · 2018-11-27
> **Reply to Reviewer 2**
>
> Thank you Reviewer 2 for your positive appraisal of our results and presentation.  As documented below, we do our best to address your questions, which have helped us improve the paper.
>
>
> Re: "The citation to Zaremba et al. in Table 1 made it seem like the perplexity result on that line of the table was directly from Zaremba et al's paper. I'd recommend removing the citation from that line to avoid confusion."
>
> We have added “similar to” in order to emphasize that we adapted and re-ran their architecture (we still use some of their code, which we believe might warrant citation; we are happy to drop it altogether if it is found confusing).
>
> Re: "One thing I would have loved to see from this paper is a comparison of modulated-plasticity LSTMs with the sota from Melis et al., 2017."
>
> We agree that, ideally, a comparison with SOTA architectures would be desirable. As explained in the response to Reviewer 1, despite all our efforts, we found the technical challenges insurmountable given our computational and engineering resources. We will keep trying to investigate such massive architectures in the future.
>
> Re: "Why were zero-sequences necessary in Experiment 1? [...] Perhaps the authors could clarify on what a confounding "time-locked scheduling strategy" would look like in this task?"
>
> The random zero-inputs make the timing of the cues unpredictable, forcing the network to be driven specifically by the stimuli - as opposed to learning a pre-programmed strategy at each given time step. This is merely a convenient choice to make the task more challenging.
>
> Re: "Why does Experiment 1 present pairs of stimuli, rather than high-dimensional individual stimuli?"
>
> Again, this simply makes the task more challenging. Non-target cues operate as distractors and having pairs of stimuli shown before each response increases the uncertainty in reward credit assignment (i.e. when receiving a reward, the network must still find out which of the two stimuli is the target).
>
> To better describe the task, we have added a schema of an episode to Figure 1. We hope this may facilitate understanding.
>
> Re: "Why is non-plastic rnn left out of Figure 2b?"
>
> As documented in Miconi et al 2018, non-plastic networks are terrible at this task. We are happy to run this experiment and include it if the reviewer finds it useful.
>
> Typos: "However, in Nature," -- no caps
> in appendix: "(see Figure A.4)" -- the figure is labeled "Figure 3""
>
> We thank the reviewer for noticing these typos and have fixed them in the text.

---

### Official Review · AnonReviewer1 · 2018-10-30
**Interesting ideas and clearly presented, but the results do not support the claims**

**Rating:** 4
**Confidence:** 4

**Review:**

This work presents Backpropamine, a neuromodulated plastic LSTM training regime. It extends previous research on differentiable Hebbian plasticity by introducing a neuromodulatory term to help gate information into the Hebbian synapse. The neuromodulatory term is placed under network control, allowing it to be time varying (and hence to be sensitive to the input, for example). Another variant proposes updating the Hebbian synapse with modulated exponential average of the Hebbian product. This average is linked to the notion of an eligibility trace, and ties into some recent biological work that shows the role of dopamine in retroactively modulating synaptic plasticity.

Overall the work is nicely motivated and clearly presented. There are some interesting ties to biological work -- in particular, to retroactive plasticity phenomena. There should be sufficient details for a reader to implement this model, thought there are some minor details missing regarding the experimental setup, which will be addressed below.

The authors test their model on three tasks: cue-award association, maze learning, and Penn Treebank (PTB). In the cue-award association task the retroactive and simple modulation networks perform well, while the non-modulated and non-plastics fail. For the maze navigation task the modulated networks perform better than the non-modulated networks, though the effect is less pronounced. Finally, on PTB the authors report improvements over baseline LSTMs.

One of the main claims of this paper is that neuromodulated plastic LSTMs...outperform standard LSTMs on a benchmark language modeling task”, and that therefore “differentiable neuromodulation of plasticity offers a powerful new framework for training neural networks”. This claim is unfortunately unfounded for a very important reason: the LSTM performance is not at all close to that which can be achieved by LSTMs in general. The authors cite such models in the appendix (Melor et al), but claim that “much larger models” are needed, potentially with other mechanisms, such as dropout. Though this may be true, these models still undermine the claim that “neuromodulated plastic LSTMs...outperform standard LSTMs on a benchmark language modeling task”. This claim is simply not true, and more care is needed in reporting the results here in the wider context of the literature. Also, I am left wondering what are considered the parameters of the models -- are only the neuromodulatory terms considered as the additional trainable parameters compared to baseline LSTMs? How are the Hebbian synapses themselves considered in this calculation? If the Hebbian synapses are not considered, then the authors need a control with matched memory-capacities to account for the extra capacity afforded by the Hebbian synapses. Given the ties between Hebbian synapses and attention (see Ba et al), an important control here could be an LSTM with Bahdanau (2014) style attention.

Finally, the style (font) of the paper does not adhere to the ICLR style template, and must be changed.

Overall, the ideas presented in the paper are intriguing, and further research down this line is encouraged. However, in its current state the work lacks sufficiently strong baselines to support the paper’s claims; thus, the merits of this approach cannot yet be properly assessed.

---

> ### Author Response · Authors · 2018-11-27
> **Reply to Reviewer 1**
>
> Thank you to Reviewer 1 for noting the clarity of our presentation and reproducibility.  We also appreciate the constructive criticism and thought that went into your review.
>
> We spent a considerable amount of time trying to fulfill the reviewer’s request to match state of the art (SOTA) on PTB. To get SOTA on PTB, we need massive architectures, which considerable computing power and experimentation at the extreme limit of what is achievable for our team. Still, we pursued two directions. First, we tried to reimplement an architecture similar to  Melis et al. 2017. However, they did not publish their code, hyperparameters, or weights, requiring re-implementing and re-training from scratch. We tried this path, but soon realized we would not be done in time (especially with a hyperparameter search).
>
> We then tried to weave neuromodulation and differentiable plasticity into the architecture and code base of Merity et al., ICLR 2018 (also tied for SOTA). However, while they could simply leverage existing PyTorch implementations of LSTMs (written in extremely fast C++), we had to re-implement LSTMs “by hand” (i.e. as a series of connected layers) in PyTorch to introduce plasticity and neuromodulation. As a result, our networks thus ran considerably slower, by more than 10x (not because our method is intrinsically slower, but just for lack of engineering optimizations on our bespoke Python implementations; we confirmed this by observing that a similar “hand-built” reimplementation of simple, non-plastic LSTMs ran similarly slower, while producing results identical to Merity et al.). These experiments are thus unfortunately still running.  For these reasons (and more provided below), we thus think it more fair (and necessary) to make such experiments the subject of a future paper.
>
> That said, we still believe the results in the current paper demonstrate the benefits of our techniques on a sizable model, and thus it would benefit the community to allow people to know about, and build upon, these new methods and results. The purpose of the present paper is to introduce a novel technique and show that it can produce an advantage in realistic settings, which we believe our PTB task confirms. Our claim is that, all other things being equal (especially the number of parameters), a neuromodulated plastic LSTM outperformed a standard LSTM on this particular benchmark task. We do **not** want to claim that our results are anywhere near SOTA. We have modified our text to avoid possible misunderstandings (see end of next-to-last paragraph in Section 4).
>
> Additionally, philosophically, If SOTA results are the bar for all papers to be accepted into conferences like ICLR, then those venues will be the exclusive domain of those with either the computation or time (i.e. large-scale resources) to dedicate to such results.  In that case, many cutting edge ideas will by necessity be excluded from the discussion, as will many research groups. Moreover, insisting on papers to be SOTA to be accepted also likely encourages p-hacking and shoddy science to game the results (even if unintentionally), reducing the quality of science our community tries to build on.
>
> Re: "Parameters of the model": All trainable parameters of the Hebbian synapses (alpha and w in Equation 1, plus the neuromodulation parameters) are included in this parameter count. To equalize the number of parameters across architectures, we reduce the number of hidden units in the plastic models in comparison to the non-plastic baseline. We have clarified this in the text.
>
> Re: "Attention": Non-trainable, homogenous plasticity can indeed be compared to a form of attention, i.e. “attending to the recent past” in the words of Ba et al. 2016. However, differentiable plasticity allows for the plasticity of each connection to be trained; as a result, different connections play different roles and it is not at all clear that the analogy with attention remains relevant (see e.g. the clever mechanisms automatically implemented by the trained plasticity connections in the image completion experiment of the Differentiable Plasticity paper, Miconi et al. 2018, sections 4.3 and S.3, which can hardly be described as simply “attention”)
>
> Re: "Style (font)": We used the template and do not see the discrepancy. Can you clarify? We are happy to fix it.

---

> > ### Comment · AnonReviewer1 · 2018-11-27
> > **quick clarification**
> >
> > Thank you for your response. I'd like to just quickly address one of your concerns regarding "SOTA".
> >
> > I agree with your philosophy here, and I don't mean to set SOTA as the required bar in any way whatsoever, and I don't believe I implied such in the review. In turn, I hope that you can appreciate that your results on PTB are not at all close to what an LSTM can achieve, nor what an LSTM could achieve 4 years ago. Indeed, there's still a 30 perplexity difference between these results and that of Zaremba 2014. Nonetheless, the number in and of itself still doesn't necessarily matter, for the reasons you state.
> >
> > However, what *does* matter is that the increase you report is on the order of ~2 perplexity, while the baseline is ~40 perplexity away from what we know LSTMs can achieve. Given that we know LSTMs can achieve 40 perplexity better, how can we be certain that the small bump you observe is indeed due to your additions, rather than some potentially unintended errors in, say, optimization or implementation of the baseline? How can we be certain that your results hold if the baseline was even a small amount closer to what we know it is capable of? With these results alone we don't necessarily know whether the effect of backpropamine will hold with better LSTMs, or whether it only works on this particular configuration of an LSTM. These points are *especially* pertinent since the baseline was performed "internally".  The desire to compare to SOTA is often to alleviate these worries more than it is to "beat a number".
> >
> > Moreover, I am not entirely convinced of the argument of requiring massive compute applies here given that a single model trained with the compute resources of 4 years ago achieves a much higher score than what is reported here. I believe there's even work done in 2012 that shows better scores using RNNs.
> >
> > Regarding the font -- please download any other ICLR paper and you will see the font is Times New Roman. Pay particular attention to the title to see the difference clearly.

---

> > > ### Author Response · Authors · 2018-12-05
> > > **Response and additional experiments**
> > >
> > > Thank you for your clarification. We appreciate the careful, highly constructive response.
> > >
> > > Regarding language modeling experiments: Following the reviewer’s suggestion, in addition to the smaller model described in the paper, we have successfully implemented plasticity in two larger architectures, including one that produces near state-of-the-art (SOTA) results.  The main result is that in all cases, neuromodulation provided a consistent improvement over non-plastic and non-modulated plastic models. Though the level of improvement is modest, these new models are large, highly optimized models, and it is interesting and supportive of our case that nevertheless simply adding neuromodulation gives a boost in all cases. In more detail:
> > >
> > > - Model 1 (medium-size architecture similar to Gal and Gharamani 2015)
> > > This model has 20 Million parameter and the authors report a test perplexity of 79.7 on test data. We ran hyperparameter search (over learning-rate, learning-rate decay and dropout rates) on this model to bring down the test perplexity to 73.96. This was the baseline model performance.
> > > Subsequently, three variants of this baseline model were evaluated: 1) with only plasticity (Miconi et al., 2018), 2) with simple neuromodulation (Equation 3 in the paper), 3) with retroactive neuromodulation (equation 4 and 5). The number of LSTM hidden units were reduced to ensure that the total number of trainable parameters remained the same as the baseline model (20 Million). In each of these variants, the same hyperparameter search was conducted as the one done on the baseline model described above. The results from a single run of each model are the following:
> > >
> > > Baseline (same model as Gal and Ghahramani 2015) with hyperparameter optimization:  73.96
> > > Baseline with plasticity, no neuromodulation (Miconi et al., 2018): 73.81
> > > Baseline with plasticity and simple neuromodulation: 73.26
> > > Retroactive neuromodulation with eligibility trace: 73.24
> > >
> > > Adding simple plasticity to the LSTM network provides a 0.15 improvement in perplexity score. However, adding either simple neuromodulation or retroactive neuromodulation to the LSTM network yields a 0.7 perplexity point improvement.
> > >
> > >
> > > - Model 2 (large-scale architecture with near-SOTA results from Merity et al. ICLR 2018)
> > > The complexity of the model (24M parameters with switching optimization algorithms) and the length of training (several days, as explained in our previous response) precluded hyperparameter tuning, so we used the same parameters as advertised on the authors’ GitHub page for all versions (the only difference being that, for all versions, we did not implement dropout in the recurrent connections because we could not integrate plasticity with the authors’ specialized code to implement it) and reduced batch size to 7 due to compute limitations. In addition, just like in model 1 above, we reduced the number of LSTM neurons in plastic models only, to ensure equal or lower total number of parameters to the baseline.
> > >
> > > With this architecture, we report the following test perplexities on PTB (single run for each model):
> > >
> > > Baseline (same as Merity et al. ICLR 2018 except for changes described above):  61.68
> > > Baseline with plasticity, no neuromodulation (Miconi et al., 2018): 61.81
> > > Baseline with plasticity and simple neuromodulation: 60.88
> > >
> > > While the 0.8 improvement of neuromodulation is modest, it still manages to produce some improvement even in a near-SOTA model, with parameters that were explicitly and carefully tuned for the non-plastic model (indeed, the importance of tuning and optimization was one of the main arguments of the paper). We believe these results confirm the results already included in the paper, showing that neuromodulation enhances the benefits of plasticity by allowing the network to control its own plasticity in real time.
> > >
> > > Re: Font issue. We have fixed the font issue and have updated our paper accordingly (it now looks similar to other ICML papers). We’ll upload the corrected version as soon as the ICLR website accepts final versions.

---

### Author Response · Authors · 2018-11-27
**General response and comments**


We thank the reviewers for their insightful comments and suggestions. We appreciate the reviewer’s agreement that the direction taken in our work is of great interest.

In response to the reviewer’s comments, the main modifications to the paper are as follows:

- We have added a figure (Figure 3) and an Appendix section (A.4) to show the dynamics of neuromodulation during performance of Task 1. This figure reveals that neuromodulation is highly dynamic and reacts to reward in complex, time-varying ways.

- We have added a schematic description of Task 1 to facilitate understanding (Figure 1, left).

- We have altered the description of Task 3 (language modeling task), and also toned down some of the description of our results.

We agree with the reviewers that, ideally, extending these results to much larger architectures capable of state-of-the-art (SOTA) results would be desirable. As explained in the response to Reviewer 1, we made every effort to implement these large architectures and augment them with plasticity and neuromodulation, given our limited resources. We regret to report that we were unable to fulfill this task in the allotted time (see response to Reviewer 1 for a description of the directions we took, and are still taking). However, we believe that our existing results (showing that plasticity and modulation improve the performance of LSTMs, **all other things being equal**, in their “signature” task of language modeling, and using non-trivial, published architectures involving millions of parameters) is in itself of great potential interest. Furthermore, we are uncomfortable with the idea that obtaining SOTA results should be a minimum bar to clear for publication of novel techniques, which might restrict innovation to a few large entities. See response to reviewer 1 for more discussion of this point.

Specific responses to individual reviewers follow.

---

### Meta-Review · Area_Chair1 · 2018-12-15

**Confidence:** 4
**Recommendation:** Accept (Poster)

**Metareview:**

The authors consider the problem of active plasticity in the mammalian brain, seen as being a means to enable lifelong learning. Building on the recent paper on differentiable plasticity, the authors propose a learnt, neuro-modulated differentiable plasticity that can be trained with gradient descent but is more flexible than fixed plasticity. The paper is clearly motivated and written, and the tasks are constructed to validate the method by demonstrating clear cases where non-modulated plasticity fails completely but where the proposed approach succeeds. On a large, general language modeling task (PTB) there is a small but consistent improvement over LSTMS. The reviewers were very split on this submission, with two reviewers focusing on the lack of large improvements on large benchmarks, and the other reviewer focusing on the novelty and success of the method on simple tasks. The AC tends to side with the positive review because of the following observations: the method is novel and potentially will have long term impact on the field, the language modeling task seems like a poor fit to demonstrate the advantages of the dynamic plasticity, so focusing on that benchmark overly much is misleading, and the paper is high-quality and interesting to the community.